# The Chain Mediating Effect of Negative Perfectionism on Procrastination: An Ego Depletion Perspective

**DOI:** 10.3390/ijerph19159355

**Published:** 2022-07-30

**Authors:** Yan Zhang, Xinwen Bai, Wanyi Yang

**Affiliations:** 1Institute of Psychology, Chinese Academy of Sciences, Beijing 100101, China; zyan1349@bjfu.edu.cn (Y.Z.); yangwy@psych.ac.cn (W.Y.); 2Department of Psychology, University of Chinese Academy of Sciences, Beijing 100049, China

**Keywords:** negative perfectionism, procrastination, ego depletion, fear of failure, cognitive flexibility, chain mediation

## Abstract

Extant research has consistently demonstrated that negative perfectionism is an important driver of procrastination. However, existing studies suffer from several salient limitations, such as an inadequacy in understanding its mediating mechanism, and the lack of an overarching theoretical framework. Accordingly, the present study adopts the ego depletion theory to investigate how and under what conditions negative perfectionism leads to procrastination. Specifically, we propose that fear of failure and ego depletion serially mediate the effect of negative perfectionism on procrastination, and that cognitive flexibility moderates this chain mediating effect. A three-wave survey consisting of 490 participants, in which negative perfectionism was measured in the first survey, fear of failure and ego depletion were measured in the second, and procrastination was measured in the last survey, lends support to all of our hypotheses. Specifically, our results indicate that (a) negative perfectionism influences procrastination through the chain mediating effect of fear of failure and ego depletion; (b) cognitive flexibility moderates the effect of fear of failure on ego depletion in that the effect is weaker when cognitive flexibility is high rather than low; and (c) cognitive flexibility moderates the chain mediating effect of negative perfectionism on procrastination, such that this chain mediation is weaker when cognitive flexibility is high rather than low. Our findings are discussed in terms of the theoretical contribution to reveal the mechanism by which negative perfectionism results in negative effects from the novel perspective of ego depletion.

## 1. Introduction

Procrastination, which is viewed as the act of unnecessarily delaying task progress or completion, and usually causes personal subjective discomfort [1], is very common in both student and general populations [2]. Though many people do not want to procrastinate because they believe it has terrible consequences, they often engage in irrational delays, making them less effective at completing tasks [2]. Studies have consistently indicated that long-term procrastination eventually turns into a habitual behavior that affects every aspect of an individual’s life, causing people to waste time and resources, and posing a threat to their physical and mental health [3,4].

Since procrastination generally has negative effects, many studies have attempted to uncover the factors that cause it. Studies have shown that some personality characteristics can influence procrastination [5,6,7]. For example, high levels of perfectionism can lead to procrastination [8]. Procrastination is positively correlated with the following aspects of perfectionism: hesitation, and fear of making mistakes. It is significantly negatively correlated with orderliness [4]. However, some studies have shown that there is a weak association between procrastination and perfectionism [2]. Further analysis has shown that this may be because procrastination is negatively correlated with conscientiousness [9], whereas perfectionism tends to correlate positively with conscientiousness [10].

Sirois et al. argue that “self-regulation theory provides a useful framework for understanding the underlying mechanisms that explain the procrastination–perfectionism relationship” [4] (p. 141). According to Sirois et al., perfectionism and procrastination are related to one another because each is the consequence of unfavorable expectancy assessments. Put differently, perfectionism or procrastination is a form of goal withdrawal or abandonment, and is commonly caused by unfavorable expectancy assessments. Although their meta-analyses reveal a small-to-medium correlation between perfectionism and procrastination, it is still unclear whether the former can cause the latter. Recent studies have indeed indicated that perfectionism can lead to procrastination via several mediators, including self-efficacy [11], a motivation to achieve [12], parenting style [13], trait anxiety [14], self-esteem [15], and depression [16]. Nevertheless, the underlying mechanism through which perfectionism results in procrastination is far from clear; more research is needed, adopting novel theoretical perspectives and taking nuanced mechanisms into account.

Baumeister’s ego depletion theory provides an overarching theoretical framework in guiding such endeavors [17,18]. Baumeister developed ego depletion theory, which predicts why an individual’s ability to regulate their behavior may fail due to limited psychological resources. The core idea is that activities requiring self-control consume a person’s resources for self-control, thereby causing ego depletion [18]. Procrastination is an avoidance behavior. In essence, it is the result of a failure of self-regulation [19]. Many maladaptive behaviors related to perfectionism can be explained by ego depletion theory [20]. For example, negative perfectionists often set high standards for themselves. They must constantly adjust their state to respond to external stress events and internal self-criticism, causing them to experience chronic stress, leading to ego depletion [21]. This paper explores the mechanism between negative perfectionism and procrastination under the theoretical framework of ego depletion.

### 1.1. Negative Perfectionism

Perfectionism is a personality trait in which an individual sets very high standards. It is accompanied by a tendency for critical self-evaluation [22], and is a complex combination of many psychological processes. Researchers have divided these processes into two types: positive perfectionism, with adaptive tendencies; and negative perfectionism, with non-adaptive tendencies [23]. Some scholars believe that “positive perfectionism” without negative emotions (e.g., anxiety) should be known as the “pursuit of excellence”. They have argued that the research on perfectionism should be controlled using clinical samples [24]. Psychological studies have tended to pay more attention to the causes, manifestations, and internal mechanisms of “negative perfectionism”.

The core traits of negative perfectionism include setting extremely high goals and standards, fearing failure, bipolar thinking (e.g., seeing imperfection as failure), and an over-reliance on markers of success and achievement in self-evaluations [25]. Based on these traits, perfectionists are often subjected to the opposing drivers of high positive motivation (i.e., a pursuit of success) and high negative motivation (i.e., an avoidance of failure). At the same time, they often have very rigid cognitive traits, and are likely to develop chronic ego depletion, leading to more negative emotions and behaviors [26,27].

### 1.2. The Ego Depletion Mechanism for Negative Perfectionism That Leads to Procrastination

According to ego depletion theory, an individual’s psychological resources are limited. If they are constantly engaged in controlling their cognition, emotions, and behavior, then their mental energy will decline until it is depleted. This can prevent them from regulating subsequent activities, and can damage their executive functioning. This is known as ego depletion [28]. Hagger et al. conducted a meta-analysis of many studies of ego depletion; they described ego depletion as the process whereby muscles experience fatigue and decline in strength after a period of activity [29]. The researchers used this theoretical framework to study the “source” and “after-effects” of ego depletion. Baumeister et al. examined ego depletion and conducted experimental field research. They argued that the causes of ego depletion are as follows: inhibited emotions or thought processes; difficulty performing mandatory tasks and making important decisions; social distress; and changes in habits [30,31]. Research into the aftereffects of ego depletion has shown that ego depletion can result in various physiological, cognitive, emotional, behavioral, and volitional (low persistence) problems [32,33].

According to ego depletion theory, the fear of failure is the main reason that negative perfectionists often experience ego depletion. This is the main trait of negative perfectionists. It is the opposite side of pursuing success when someone is motivated to achieve, and it is an avoidance behavior [34]. Other traits of negative perfectionism are also based on the fear of failure, including “hesitation, excessive carefulness, and excessive planning”. Studies have indicated that when negative perfectionists pursue challenging goals, the fear of failure can encourage them to work hard, but it also causes high levels of anxiety [35]. It can also damage their motivation and their psychological resilience, resulting in ego depletion [36,37]. Therefore, the fear of failure is probably a dominant factor influencing perfectionists’ ego depletion. In addition, some studies have shown that the fear of failure is the primary factor among some people who procrastinate [38]. As a person’s fear of failure keeps increasing, it causes negative emotions, such as a lack of confidence. These negative emotions can affect behavior, as people do not dare to try in their endeavors. Therefore, the fear of failure is a high-risk loss source in promoting negative perfectionists to present procrastination.

In addition, different individuals consume their psychological energy at different rates. Salmon et al. refer to the rate of ego depletion as “loss sensitivity”; the higher the loss sensitivity, the faster the loss of psychological energy [39]. Since negative perfectionists set high standards, they are very likely to face highly difficult and mandatory tasks externally. They also suffer from complex emotional conflict. Therefore, negative perfectionists are highly susceptible to ego depletion. People who are suffering from ego depletion are unable to control themselves and their environment, because self-regulation is resource-demanding. Consequently, they often find it difficult to initiate an action or to regulate themselves toward task completion. As a result, individuals in a resource-depleted state tend to postpone the initiation, advancement, or completion of a task, which unavoidably causes procrastination [40]. Therefore, procrastination is often an aftereffect of ego depletion. In other words, ego depletion can act as the mechanism of transferring the effect of negative perfectionism on procrastination.

In summary, ego depletion theory provides us with an overarching framework to reveal the effect of negative perfectionism on procrastination. In contrast to most studies, this paper explores the variables that affect the causes and effects of ego depletion. It uses the connected overall framework of sources and aftereffects to explore the maladaptive behaviors caused by certain personality traits.

**Hypothesis** **1.**
*Negative perfectionism is positively correlated with procrastination.*


**Hypothesis** **2.**
*Negative perfectionism affects procrastination through the chain mediating effect of the fear of failure and ego depletion.*


### 1.3. The Moderating Effect of Cognitive Flexibility

Based on ego depletion theory, this study further speculates that cognitive flexibility is an important boundary condition determining the extent to which negative perfectionists’ fear of failure contributes to ego depletion. Cognitive flexibility is a core component of a person’s executive functioning. It refers to their ability to adjust their thinking, overcome inherent patterns of thought, and adapt to new scenarios in a changing environment [41]. It has three core dimensions: (1) optionality, i.e., being able to make effective selections in any scenario; (2) adaptability, i.e., being able to adapt flexibly to specific environments; and (3) controllability, i.e., being effective when taking actions [42]. Hayatbini et al. found that perfectionism is negatively correlated with cognitive flexibility [43]. In addition, some studies have shown that poor cognitive flexibility can be an important component of obsessive–compulsive disorder [44]. This shows that many adverse adaptations of perfectionism are related to defects in cognitive flexibility.

This study holds that these three core characteristics of cognitive flexibility can alleviate the irrational beliefs that perfectionists suffer from due to their fear of failure, thereby reducing ego depletion. Firstly, optionality challenges the rigid thinking of negative perfectionism; cognitive flexibility can encourage perfectionists to realize that there are multiple explanations for the causes of specific events and behaviors [43]. This allows perfectionists to realize that “everything can fail, not everything happens at once, and things may have to be accomplished in stages”, and can greatly reduce people’s fear of failure and the anxiety that accompanies it. To some extent, cognitive flexibility can be a mediator, allowing people to maintain high standards while still being relaxed in the pursuit of success. Secondly, in terms of adaptability, perfectionists tend to respond to maladaptive behaviors rather than adjust their unrealistic standards. Cognitive flexibility can encourage cognitive reconstruction, enabling individuals to consider multiple ideas and replace compulsive coping mechanisms with more flexible and well-adapted cognitive styles [44]. Therefore, cognitive flexibility tends to make perfectionists more adaptable, helping them to adopt multiple ways of coping with difficulties. Finally, in terms of controllability, individuals with high cognitive flexibility “can flexibly switch cognitive sets and inhibit habitual response patterns” [45]. Therefore, cognitive flexibility supplements a person’s resources of self-control, allowing them to overcome their fear of failure and minimize ego depletion to a certain extent.

In summary, negative perfectionists’ fear of failure often originates from the binary idea that “imperfection means failure”. Individuals with low cognitive flexibility find it difficult to filter threat that is irrelevant to a task. They are often unable to adjust to new information quickly. They resist change, and insist on unreasonable forms of cognition and responses [46]. Conversely, individuals with high cognitive flexibility can break with rigid patterns of thinking, thereby reducing the loss of ego resources that is caused by the fear of failure. Therefore, the following hypothesis is proposed:

**Hypothesis** **3.**
*Cognitive flexibility moderates the effect of the fear of failure on ego depletion. Specifically, the fear of failure yields a weaker effect on ego depletion when cognitive flexibility is high rather than low.*


The above hypotheses constitute a moderated mediation model. Specifically, negative perfectionism leads to procrastination through the serial transmitting mechanism of the fear of failure and ego depletion (Hypothesis 2). However, cognitive flexibility mitigates the effect of the fear of failure on ego resources (Hypothesis 3), thereby resulting in a reduced, indirect effect of negative perfectionism on procrastination. In other words, the chain mediating effect of negative perfectionism on procrastination is reduced for individuals with high levels of cognitive flexibility. Thus, the following hypothesis is proposed:

**Hypothesis** **4.**
*Cognitive flexibility moderates the chain mediating effect of negative perfectionism on procrastination through the fear of failure and ego depletion. Specifically, this chain mediating effect is weaker for those with higher levels of cognitive flexibility than for their counterparts with lower levels of cognitive flexibility.*


Figure 1 depicts our overall theoretical model, in which fear of failure and ego depletion serially transmit the effect of negative perfectionism on procrastination, and cognitive flexibility serves as the boundary condition for this chain mediation process.

## 2. Materials and Methods

### 2.1. Samples and Procedures

In this study, questionnaires were distributed to college students in Beijing using Credamo (www.credamo.com, accessed on 16 March 2022)—a reliable Chinese online survey platform. To avoid the common methodological biases, data were collected using a three-wave survey strategy with a lag of one month between each survey wave. In the first survey, participants’ negative perfectionism (the independent variable) and cognitive flexibility (the moderator), along with their demographics, were measured. Information about participants’ backgrounds was also recorded. In total, 600 participants gave valid answers to the survey. In the second survey, two mediators (i.e., fear of failure and ego depletion) were measured. Only those who had completed the first survey (*n* = 600) were invited to participate in the second survey, of whom 516 gave valid answers to the survey, resulting in a valid response rate of 86%. Similarly, only those who had completed the second survey (*n* = 516) were invited to participate in the last survey, which aimed to measure the dependent variable (i.e., procrastination). Among them, 490 participants returned valid answers to the survey, resulting in a valid response rate of 95%. Therefore, the present study relied on the survey data of the final sample (*n* = 490) to test all of the hypotheses. The participants’ mean age was 24.08 years (ranging from 18 to 38 years, with SD = 4.17). Table 1 presents the demographic characteristics of the final sample.

### 2.2. Measures

Unless stated otherwise, all variables were measured by asking the participants to indicate to what extent each item could describe themselves on a 5-point scale (1 = very inaccurate, 5 = very accurate).

**Negative perfectionism**. Negative perfectionism was measured using Zi’s Negative Perfectionism Questionnaire [22,25]. Zi developed this scale by conducting in-depth interviews to capture perfectionists’ negative psychological features in the Chinese context, and by combining several items of Frost’s Multidimensional Perfectionism Scale [23,26]. This scale consists of 38 items that belong to one of five sub-dimensions: 10 items on “hesitancy”, 7 items on “the fear of failure”, 9 items on “excessive caution and carefulness”, 6 items on “excessive planning and control”, and 6 items on “extremely high goals and standards”. Higher scores indicated a stronger tendency of negative perfectionism. In this study, the internal consistency coefficient was 0.90.

**Fear of failure**. The Chinese version of Conroy’s Assessment Scale of the Fear of Failure, revised by Sun (2007), was adopted [36,39]. This scale consists of a total of 20 items that belong to one of 5 sub-dimensions: 6 items on “the fear of experiencing shame and embarrassment”, 4 items on “the fear of a lower self-evaluation”, 3 items on “feeling confused about the future”, 4 items on “the fear of a lower social value”, and 3 items on “disappointing significant others”. Higher scores indicated a higher level of fear of failure. The internal consistency coefficient for the scale was 0.96.

**Ego depletion**. The Chinese version of Nes et al.’s Self-Regulatory Fatigue Scale, revised by Wang et al., was adopted [47,48]. In total, 18 items were included, divided into 3 categories: cognition, emotion, and behavior. Participants were asked to what extent they agreed with each description on a 5-point scale, ranging from 1 (strongly disagree) to 5 (strongly agree). Higher scores indicated a worse state of ego depletion. The internal consistency coefficient for the scale was 0.93.

**Procrastination**. The revised Chinese version of Lay’s General Procrastination Scale was used [49]. The scale has a single category with 20 items. Each item was scored using a 5-point scale ranging from 1 (very inconsistent) to 5 (very consistent). Higher scores indicated a higher degree of procrastination. The internal consistency coefficient for the scale was 0.94.

**Cognitive flexibility**. The revised Chinese version compiled by Dennis was adopted [42]. In total, 20 items were included, which were divided into 2 categories: optionality and controllability. The items were scored using a 5-point scale ranging from 1 (never) to 5 (always). Six items of reverse scoring were included. With the reverse scoring, the higher the total score, the higher the cognitive flexibility. The internal consistency coefficient for this scale was 0.90.

**Demographic variables**. As presented in Table 1, participants’ demographic characteristics—including gender, age, whether they were an only child, whether they were from a single-parent family, grade, and major—were collected.

### 2.3. Analytical Strategies

We first used a series of hierarchical regression analyses to test all of our hypotheses, and further constructed bias-corrected confidence intervals to test hypotheses involving indirect effects (i.e., Hypotheses 2 and 4) using PROCESS [50].

## 3. Results

### 3.1. Test of Common Method Biases

Although the questionnaire data were collected at three different times, all of the data were self-reported. To test whether there were common methodological biases, Harman’s single-factor test was used to analyze all of the items of the five key variables of this study [51]. The amount of variation explained by the first factor was 24.97%, which is lower than 40%. This indicates that the common methodological biases in this study had no significant effect.

### 3.2. Descriptive Statistics

As indicated in Table 2, negative perfectionism was positively and significantly correlated with fear of failure (*r* = 0.44, *p* < 0.001), ego depletion (*r* = 0.25, *p* < 0.001), and procrastination (*r* = 0.18, *p* < 0.001). Moreover, both fear of failure (*r* = 0.44, *p* < 0.001) and ego depletion (*r* = 0.67, *p* < 0.001) were positively and significantly correlated with procrastination. These results provide preliminary support for the hypothesized chain mediation.

### 3.3. Hypothesis Testing

Next, we conducted a series of hierarchical regression analyses to formally test all of the hypotheses. Consistent with the chain mediation model depicted in Figure 1, negative perfectionism was always the independent variable (IV) and procrastination was always the dependent variable (DV) in all regression models; either fear of failure or ego depletion could be the IV or DV depending on what DV the specific regression analysis was targeting. In all models of the hierarchical regression analyses, five demographic variables were included in the first step as the control variables. The results are shown in Table 3. It can be seen that negative perfectionism positively and significantly predicted procrastination (Model 8: *β* = 0.15, *t* = 3.24, *p* < 0.01), lending support to Hypothesis 1.

Hypothesis 2 predicted that negative perfectionism would influence procrastination through the chain mediation of fear of failure and ego depletion. The results showed that when fear of failure was included in the regression equation (i.e., Model 9), it was positively and significantly associated with procrastination (*β* = 0.42, *t* = 9.24, *p* < 0.001), but negative perfectionism could no longer significantly predict procrastination (*β* = −0.03, *t* = 0.74, *p* = 0.46). When ego depletion was further included in the final regression model (i.e., Model 10), it was positively and significantly associated with procrastination (*β* = 0.68, *t* = 14.67, *p* < 0.001); however, neither negative perfectionism (*β* = 0.005, *t* = 0.14, *p* = 0.89) nor fear of failure (*β* = −0.01, *t* = 0.26, *p* = 0.79) could significantly predict procrastination. In addition, negative perfectionism could positively and significantly predict fear of failure (Model 2: *β* = 0.42, *t* = 10.29, *p* < 0.001) and ego depletion (Model 4: *β* = 0.22, *t* = 5.06, *p* < 0.001), while fear of failure could positively and significantly predict ego depletion (Model 5: *β* = 0.66, *t* = 17.22, *p* < 0.001). All of these results provide strong evidence for the chain mediation, as stated in Hypothesis 2.

It was also hypothesized that cognitive flexibility would moderate the effect of fear of failure on ego depletion (Hypothesis 3). As shown by Model 6 in Table 3, the interaction between cognitive flexibility and fear of failure could significantly predict ego depletion (*β* = −0.17, *t* = −5.64, *p* < 0.001). To further examine whether the direction of the interaction was consistent with the hypothesis, we conducted simple slope tests and graphically showed the interaction patterns (see Figure 2). The simple slope analyses showed that when cognitive flexibility was higher (+1 standard deviation), although fear of failure still positively predicted ego depletion (b = 0.28, *t* = 10.45, *p* < 0.001), the simple slope was smaller compared with when the level of cognitive flexibility was lower (−1 standard deviation; b = 0.47, *t* = 17.16, *p* < 0.001). Put differently, fear of failure yielded a weaker effect on ego depletion when cognitive flexibility was high rather than low. Taken together, Hypothesis 3 was supported.

We employed Hayes’s PROCESS macro [50] to test the chain mediating effect (Hypothesis 2) and the moderated chain mediating effect (Hypothesis 4). Specifically, we relied on PROCESS to construct bias-corrected, bootstrapped confidence intervals with 5000 iterations to test these two hypotheses involving indirect effects. The indirect effect is significant if its 95% confidence interval (CI) excludes zero. The results are shown in Table 4. As can be seen, the chain mediating effect (i.e., the indirect effect of negative perfectionism → fear of failure → ego depletion → procrastination) was positive and significant, since its CI excluded zero (indirect effect = 0.29, 95% CI = [0.23, 0.35]), supporting Hypothesis 2. Furthermore, it can also be seen from Table 4 that the two simple mediating effects were not significant, given that the CI for each indirect effect included zero (negative perfectionism → fear of failure → procrastination: indirect effect = −0.01, 95% CI = [−0.06, 0.03]; negative perfectionism → ego depletion → procrastination: indirect effect = −0.04, 95% CI = [−0.10, 0.01]). These results provide further evidence for the hypothesized chain mediating effect.

Hypothesis 4 predicted that cognitive flexibility would moderate the chain mediating effect of negative perfectionism on procrastination. Accordingly, we constructed a moderated chain mediation model based on our research model (see Figure 1). Results obtained from the PROCESS macro indicated that the chain indirect effect was significant when the cognitive flexibility level was high (indirect effect = 0.17, 95% CI = [0.12, 0.23]) or low (indirect effect = 0.32, 95% CI = [0.26, 0.39]). However, there was a significant difference between the two indirect effects (difference = −0.15, 95% CI = [−0.23, −0.08]), indicating that the former was significantly smaller than the latter. Thus, Hypothesis 4 was supported.

To further determine whether cognitive flexibility only played a moderating role between fear of failure and ego depletion, we conducted a series of supplementary analyses. We first explored whether cognitive flexibility would moderate the effect of negative perfectionism on the fear of failure (i.e., the first stage of the chain mediation model). The results of hierarchical regression analyses indicated that the interaction between cognitive flexibility and negative perfectionism could not significantly predict fear of failure (*β* = 0.10, *t* = 0.69, *p* = 0.49). Next, we examined whether cognitive flexibility would moderate the effect of ego depletion on procrastination (i.e., the third stage of our chain mediation model). Again, the interaction between cognitive flexibility and ego depletion could not significantly predict procrastination (*β* = −0.08, *t* = −0.13, *p* = 0.19). Finally, using the same procedure concerning the testing of Hypothesis 2, we employed Hayes’s PROCESS macro [50] to test whether the moderated chain mediating effect was significant if cognitive flexibility moderated the path of stage 1 (i.e., negative perfectionism → fear of failure) or stage 3 (i.e., ego depletion → procrastination). The results showed that there was no significant difference between the two indirect effects when cognitive flexibility moderated stage 1 (difference = 0.03, 95% CI = [−0.05, 0.15]) or stage 3 (difference = −0.03, 95% CI = [−0.10, 0.04]). Taken together, these results indicate that, as expected, cognitive flexibility could only moderate the chain mediating effect at stage 2. Therefore, the results of our supplementary analyses provide further support for Hypothesis 4.

## 4. Discussion

Based on the data from a three-wave survey, this study shows that ego depletion is an internal mechanism that makes it more likely that negative perfectionism will lead to procrastination. This suggests that ego depletion theory is a good theoretical framework for explaining the relationship between negative perfectionism and procrastination. The analysis of the mediating effects shows that negative perfectionism leads to procrastination through (and only through) the chain mediating effect of the fear of failure and ego depletion. The analysis also shows that the chain mediating effect of negative perfectionism leading to procrastination is regulated by an individual’s cognitive flexibility. When a person’s cognitive flexibility is higher, the degree of procrastination caused by their negative perfectionism is lower. This is because cognitive flexibility alleviates the depletion of ego resources by the fear of failure.

Consistent with results of Sirois et al.’s recent meta-analysis [4], our study shows that perfectionism is positively and significantly related to procrastination, yet the correlation magnitude (*r* = 0.18) is somewhat small. At first glance, this might imply that negative perfectionism is not a salient or strong predictor of procrastination. Nevertheless, our results of the chain mediation model add more nuance to the bourgeoning discussions of the procrastination–perfectionism association. Our study indicates that negative perfectionism can lead to procrastination through (and only through) the chain mediating mechanism of the fear of failure and ego depletion. Therefore, distant though their association might seem, this definitely does not mean that such association is unimportant or can be ignored. Rather, more research is needed to reveal the underlying mechanisms.

### 4.1. Theoretical Contributions

The most important theoretical contribution of this study lies in its introduction of ego depletion theory as a guiding framework. This expands the understanding of how perfectionism leads to negative behaviors, and provides a new perspective on how procrastination occurs. The “depletion view” and “preservation view” of ego depletion [52] can help us to understand how negative perfectionism can lead to procrastination. The first stage of the process involves the depletion of ego resources. Negative perfectionists often have a fear of failure. When this kind of “situational fear of failure” is triggered, such individuals find that they have to pursue extremely high standards while bearing an emotional burden. As a result, they can suffer an emotional crisis involving “threat, fear, anxiety, shame, fear of mistakes, and being forced to make an effort”. This can lead to ego depletion. The second stage involves the motivation to conserve resources once the ego has been depleted. Negative perfectionism leads to procrastination as a result of ego depletion. There are two reasons for this: First, an individual’s mental energy declines until the state of depletion, and then the procrastination behavior with decreasing goal dependence is generated, which is the result of a failure of self-regulation. Second, once an individual becomes aware of their lack of energy, they subconsciously introduce protective mechanisms to conserve their remaining energy and avoid depleting their ego. Negative perfectionists are very likely to choose the strategy of temporary ego evasion (procrastination).

The second theoretical contribution of this study is that it reveals that cognitive flexibility is an important boundary condition for negative perfectionism leading to procrastination through the fear of failure and ego depletion. This provides a theoretical reference point for researchers trying to understand the regulatory effects of cognitive flexibility. Firstly, this study shows that cognitive flexibility plays an important role in the “recognition and regulation” of emotions [53]. Specifically, individuals with high cognitive flexibility are more likely to perceive their emotional state clearly and adjust it. Thus, cognitive flexibility can regulate a person’s fear of failure, reducing their emotional friction. Secondly, this paper shows that high cognitive flexibility can overcome the irrational beliefs and negative responses that are common among negative perfectionists. The higher a person’s cognitive flexibility, the stronger their self-control, the easier they find it to adapt to changes, and the more aware they are of different possible choices. These aspects can encourage individuals to use cognitive strategies, break away from rigid patterns of thought, and avoid compulsive coping mechanisms. They can also help individuals to adopt more “efficient and energy-saving” behavioral strategies. Meanwhile, as cognitive flexibility is at the core of executive functioning, it can allow people to have a higher level of self-control. In summary, the higher an individual’s cognitive flexibility, the less likely they are to procrastinate.

### 4.2. Practical Implications

This study has implications for clinical interventions endeavoring to reduce the effects of negative perfectionism and prevent procrastination. Negative perfectionists do not have the trait of procrastination; in essence, their procrastination is avoidant procrastination. The main factor that promotes avoidant procrastination is the “fear of failure”. Therefore, a key direction in clinical practice should be to help perfectionists overcome their fear of failure, build their self-confidence, and prevent the development of avoidance behaviors.

The results of this study show that cognitive flexibility can significantly reduce the impact of negative perfectionism leading to procrastination. This helps to clarify the best ways to reduce negative perfectionists’ tendency to procrastinate. It is difficult for procrastinators to give up their high standards, because they are formed by deep-seated thoughts and values. However, they can use cognitive flexibility to help them pursue success and avoid the fear of failure. This can transform them from being passive to pursuing high standards actively, which can help them to transition from negative perfectionism to positive perfectionism.

### 4.3. Limitations and Future Directions

This study has some limitations. Firstly, although we conducted the three-wave survey to measure independent, mediating, and dependent variables at different times, as recommended for mitigating common methodological biases [5], our study was still correlational by nature, and was not able to reveal causal relationships between variables. In follow-up studies, a longitudinal study design or experimental design could be used, since these would help to reveal the causal relationships between the variables. Secondly, all of the subjects in this study were college students. Future studies could examine whether the results of this study can be extended to other groups. Thirdly, the chain mediation hypothesis focused on the fear of failure as a source of ego depletion. Other consequences of negative perfectionism, such as self-esteem and self-efficacy [11,15], might also lead to ego depletion. We encourage future studies to systematically explore other mechanisms through which negative perfectionism can lead to negative consequences.

## 5. Conclusions

Ego depletion theory provides a new theoretical perspective in revealing the mechanisms by which negative perfectionism leads to procrastination. The results of this study indicate that negative perfectionism leads to procrastination through (and only through) the chain mediating effect of the fear of failure and ego depletion. This has important implications for negative perfectionists looking to overcome procrastination. In addition, this study also shows that cognitive flexibility serves as an important boundary condition in mitigating the negative effect of negative perfectionism on procrastination through the fear of failure and ego depletion. Therefore, improving cognitive flexibility could help negative perfectionists to reduce procrastination.

## Figures and Tables

**Figure 1 ijerph-19-09355-f001:**
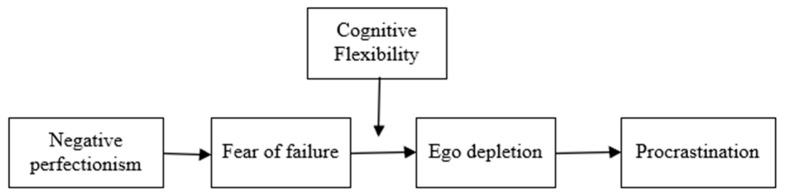
Theoretical model.

**Figure 2 ijerph-19-09355-f002:**
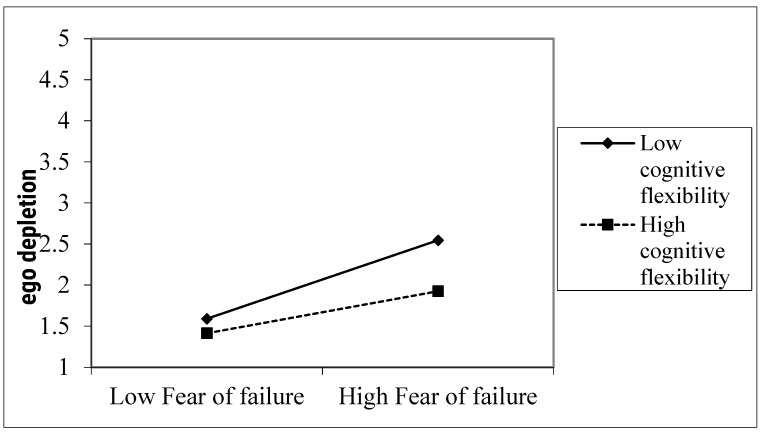
Interaction of the fear of failure and cognitive flexibility on ego depletion.

**Table 1 ijerph-19-09355-t001:** Demographic characteristics of the final sample (*n* = 490).

Variable	Category	Frequency	Percentage (%)
Gender	Male	190	38.8
	Female	300	61.2
Grade	Undergraduate	353	72.0
	Postgraduate	137	28.0
Major	Humanities and social sciences	207	42.2
	Science and engineering	283	57.8
From a single-parent family?	Yes	25	5.1
	No	465	94.9
Being the only child?	Yes	300	61.2
	No	190	38.8

**Table 2 ijerph-19-09355-t002:** Mean values, standard deviations, and correlation coefficients for all variables.

Variables	*M* (*SD*)	1	2	3	4	5	6	7	8	9	10	11
1. Gender	0.61 (0.49)	-										
2. Grade	0.28 (0.45)	0.04	-									
3. Age	24.08 (4.17)	0.04	0.62 ***	-								
4. Major	0.58 (0.49)	−0.36 ***	−0.08	−0.07	-							
5. Single-parent family	0.95 (0.22)	0.03	−0.02	−0.04	0.05	-						
6. Only child	0.39 (0.49)	−0.03	−0.04	−0.05	−0.02	0.05	-					
7. Negative perfectionism	3.44 (0.48)	0.11 *	0.09 *	0.07	−0.08	0.15 **	−0.11 *	(0.90)				
8. Fear of failure	2.53 (0.96)	0.03	0.12 *	0.09 *	−0.07	0.01	−0.22 **	0.44 ***	(0.96)			
9. Ego depletion	1.96 (0.68)	0.05	0.06	0.04	−0.04	0.03	−0.29 **	0.25 ***	0.66 ***	(0.93)		
10. Procrastination	1.99 (0.66)	0.04	0.06	−0.01	0.02	0.10 *	−0.23 **	0.18 ***	0.44 ***	0.67 ***	(0.94)	
11. Cognitive flexibility	4.06 (0.47)	0.06	−0.01	−0.04	−0.01	−0.04	0.14 **	−0.02	−0.28 ***	−0.49 ***	0.49 ***	(0.90)

Note: *n* = 490. *** *p* < 0.001, ** *p* < 0.01, * *p* < 0.05. Gender: 0 = male, 1 = female. Grade: 0 = undergraduate, 1 = postgraduate. Major: 0 = humanities and social sciences, 1 = science and engineering. Single-parent family: 0 = no, 1 = yes. Only child: 0 = no, 1 = yes. The number in parentheses on the diagonal is the internal consistency coefficient α measured for each variable.

**Table 3 ijerph-19-09355-t003:** Results of regression analyses.

Predictor Variable	DV: Fear of Failure		DV: Ego Depletion				DV: Procrastination			
	Model 1	Model 2	Model 3	Model 4	Model 5	Model 6	Model 7	Model 8	Model 9	Model 10
Control variables										
Gender	−0.01	−0.04	0.03	0.01	0.04	0.04	0.03	0.22	0.04	0.01
Age	0.04	0.03	0.00	−0.00	−0.02	−0.04	−0.05	−0.08	−0.09	−0.07
Grade	0.07	0.04	0.04	0.03	0.00	0.01	0.10	0.09	0.07	0.07
Major	−0.07	−0.04	−0.03	−0.02	0.01	0.01	0.03	0.03	0.05	0.04
Single-parent family	0.02	−0.04	0.04	0.01	0.03	0.02	0.11 *	0.09	0.10 *	0.08 *
Only child	−0.22 ***	−0.17 ***	−0.29 ***	−0.27 ***	−0.16 ***	−0.11 ***	−0.23 ***	−0.21 ***	−0.14 **	−0.04
Independent variable										
Negative perfectionism		0.42 ***		0.22 ***	−0.06	−0.02		0.15 **	−0.03	0.005
Mediators										
Fear of failure					0.66 ***	0.56 ***			0.42 ***	−0.01
Ego depletion										0.68 ***
Moderator and interaction terms										
Cognitive flexibility						−0.30 ***				
Cognitive flexibility × fear of failure						−0.17 ***				
*R^2^*	0.05 ***	0.22 ***	0.08 ***	0.13 ***	0.46 ***	0.58 ***	0.06 ***	0.08 **	0.21 ***	0.46 ***
*F*	5.39 ***	20.73 ***	8.01 ***	10.88 ***	52.37 ***	67.74 ***	6.24 ***	6.99 ***	18.04 ***	47.00 ***
*ΔR^2^*		0.17 ***		0.05 ***	0.33 ***	0.12 ***		0.02 **	0.13 ***	0.24 ***
*ΔF*		105.73 ***		25.63 ***	296.35 ***	69.17 ***		10.65 **	86.74 ***	212.59 ***

Note: *n* = 490. *** *p* < 0.001, ** *p* < 0.01, * *p* < 0.05. DV = dependent variable. Gender: 0 = male, 1 = female. Grade: 0 = undergraduate, 1 = postgraduate. Major: 0 = humanities and social sciences, 1 = science and engineering. Single-parent family: 0 = no, 1 = yes. Only child: 0 = no, 1 = yes.

**Table 4 ijerph-19-09355-t004:** Analyses of the mediating effect.

Mediating Path	Estimates	95% CI
Mediating effect		
Chain mediating effect: Negative perfectionism → fear of failure → ego depletion → procrastination	0.29	[0.23, 0.35]
Simple mediating effect 1: Negative perfectionism → fear of failure → procrastination	−0.01	[−0.06, 0.03]
Simple mediating effect 2: Negative perfectionism → ego depletion → procrastination	−0.04	[−0.10, 0.01]
Total indirect effect	0.23	[0.16, 0.31]
Direct effect	0.03	[−0.05, 0.10]
Total effect	0.26	[0.16, 0.35]
Moderated chain mediating effect		
Cognitive flexibility: High (+1 SD)	0.17	[0.12, 0.23]
Cognitive flexibility: Low (−1 SD)	0.32	[0.26, 0.39]
Difference between high versus low levels of cognitive flexibility	−0.15	[−0.23, −0.08]

## Data Availability

Data are available upon request from the corresponding author. The data are not publicly available due to privacy or ethical restrictions.

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
