# Peer review of "The Chain Mediating Effect of Negative Perfectionism on Procrastination: An Ego Depletion Perspective"

_ijerph, 2022, doi:10.3390/ijerph19159355_

Round 1

Reviewer 1 Report

The manuscript is written well and exceptionally carefully.
Nevertheless, there is a lack of some key components of scientific article which are necessary.
Firstly, the authors should add a statistics description in the method section. It is an essential element of methodology and should be not omitted.
Secondly: Please, expand the discussion by comparing your results to this obtained by other researchers.
Some less critical issues:
Line 39, 40: please, separate citations from „procrastination”. Make this change in the following sections of the main text.
51-59: please, add literature
Line 358: dot is probably unnecessary. Please verify.

After changes, I highly recommend publishing this article.

Reviewer 2 Report

The authors should proofread the manuscript carefully and revise the writing logic.

Comments to the Author

Major points

1. Introduction

-The relationship between negative perfectionism and procrastination

The authors proposed that “Procrastination is generally regarded as a typical sign of negative perfectionism” and explored the relationship between negative perfectionism and procrastination. My question is “Does procrastination belong to negative perfectionism?”

- The mediating role of the fear of failure

There is a similar question: because one of aspects of negative perfectionism is “the fear of failure”, what is the difference between the aspect of negative perfectionism (i.e., the fear of failure) and the mediator variable (i.e., the fear of failure)?

-The mediating role of the ego depletion

The authors proposed that “ego depletion can cause negative perfectionism, which leads to procrastination” (see Line 112-113, Page 3). Moreover, in the part, the authors did not describe or discuss the relation between ego depletion and negative perfectionism. It is confusing that negative perfectionism could be a mediator between ego depletion and procrastination. This is in contradiction with the results of this study.

3. Method

The authors conducted a three-wave longitudinal survey to investigate their hypotheses. Data was collected with a one-month interval from internet. It is a benefit to use longitudinal approach in causal inference (even so, the causality is much weaker than laboratory studies).

-Demographic variables: The authors should report the age of the participants. Please check the demographic variable (i.e., age, see Line 256–258, Page 5).

4. Results:

-Why not control age in the regression analysis (see Table 3)? Please check whether the results could be identical when control the age.

-Please check the accuracy and consistency of writing in paper (e.g., β = .15, see Line 284, Page 7) and Table (β = 0.15, see Table 3).

- Tables and figures: Please check the formats according to the author guidance.

-Table 2: You could add the information of the demographic variable (i.e., age, M and SD). Please check the notes “p” (Line 273-274, Page 7).

-Table 3: what is the difference between “/” and blanks?

5. Discussion:

Please check the accuracy of writing.

-For example, “Based on the data from the three questionnaires (see Line 342, Page 10)” was false. The authors conducted three-wave data survey rather than three questionnaires.

-Most of the discussion was well written. But some statement should be considered carefully. For example, it is not evidenced that negative perfectionism leads to procrastination as a result of ego depletion. The authors cannot make such conclusion, because the results showed that ego depletion was only mediated the relation between the fear of failure and procrastination in the chain mediating effect (see Table 4).

-More references in discussion are needed especially in theoretical contributions.

6.Conclusion

Please check the conclusion.

-Some statement should be considered carefully. For example, the authors proposed that “The results of this study indicate that the fear of failure plays a key role in negative perfectionism, which leads to procrastination and ego depletion.” However, I cannot make such conclusion solely from the study, because the study did not confirm that the fear of failure could be a mediator in the relation between negative perfectionism and procrastination neither the theoretical arguments nor the results.

7.References

Please check the accuracy of reference style (e.g., reference 32, see Line 512-513, Page 13). Moreover, the usage of dashes was totally chaos.

Proofing: the authors should proofread the manuscript carefully and revise the writing logic.

Reviewer 3 Report

This manuscript presents a correlational study exploring the mechanism that relates negative perfectionism to procrastination. Theoretically, the authors argue for the novelty of framing the study of such a mechanism in a larger psychological theory – in this case, the “ego depletion theory”. Self-reported information was collected in a sample of 490 young adults.

Overall, the study was well conducted, and the results are interesting, with relevant theoretical and practical implications. I have no special structural comments on the manuscript but only some small comments that I think, if kindly considered by the authors, may improve the readability of the text.

First, I agree with the authors that exploring isolated mediating factors is too erratic and, considering the risk of false positives and publication bias for significant results, it will not contribute to the systematic understanding of the process of procrastination. However, the authors are not totally fair when they say that a theoretical framework is lacking in the literature about this relationship. For instance, in their meta-analysis, Sirois et al. (2017) stated that “self-regulation theory provides a useful framework for understanding the underlying mechanisms that explain the perfectionism-procrastination relationship”. Also, Xie, Yang & Chen (2018; Social Behavior and Personality) suggested that temporal motivational theory can be used as a framework to support the mediational role of self-efficacy in this relationship. So, I suggest the authors soften their statement concerning the absence of theoretical guidance for testing putative mediating variables. After all, the theoretical proposal of this manuscript is as legitimate as those previous ones that I have mentioned.

Second, the second limitation pointed to existing studies (lines 55-59) is not totally clear, at least to me. Why do the authors state that “there has been no in-depth research into the extensive mediating variables and moderator variables”? Is the present study an example of “in-depth research? Why? I believe the readers are expecting to grasp a clear understanding of this statement.

To me, Hypotheses 3 and 4 seem rather redundant. While H3 is clearer because it specifies the focus of moderation, H4 is fuzzy (it does not explicitly state where the moderator exerts its effects). Furthermore, being not specific, H4 is not really tested. So, I suggest that H4 became the generic moderation hypothesis (H3) and then H3 became the specific moderation hypothesis (H3a) that is truly tested.

Still concerning the test of the chain mediation hypothesis, the authors tested for the two short mediational chains, founding null indirect effects (lines 321-325). I suggest that the existence of a direct effect should also be tested (the direct path perfectionism -> procrastination), to make sure that only the complete chain is the reliable mechanism of influence.

Since the total effect of perfectionism on procrastination is small in this sample (r = .18), is it possible to know the percentage of the indirect effect compared to this total effect? Is there any margin for additional mediational chains?

In lines 381-383, the authors state that a “person who possesses the three aspects of cognitive flexibility (optionality, adaptability, and controllability) can overcome the irrational beliefs and negative responses that are common among perfectionists”. However, this is speculation – no analysis using cognitive flexibility subscales was done. Please, adjust.

In the Limitations section, why do the authors say that there were problems with questionnaires? What problems were these and what were their consequences?

In the Discussion, I ask the authors to address the fact that the total effect of Negative Perfectionism on Procrastination is pretty small, although statistically significant (r = 0.18; R2 = 3.2%; in regression model 8, it reaches only R2 = 2%). So, the overall mediated effect is necessarily small (less or equal to 2%), and, as a consequence, negative perfectionism does not seem to be a particularly relevant predictor of procrastination. This correlation magnitude (r = .18) is somehow similar to the one reported in the 2017 meta-analysis (r ~.23).

Minor

I suggest the authors provide information concerning participants' age (in Table 1, for instance).

Id correlations were used to describe the relationship between sociodemographic and psychological variables (as suggested by Table 2’s legend), I ask the authors to indicate the numeric codes for those dichotomic variables (I believe the codes were female =1; postgraduate = 1, etc). This is crucial to understanding biserial correlations presented in Table 2.

Please, check the figures presented on the cells corresponding to Major vs. Gender correlation (62.94) or Major vs. Grade correlation (2.74), and so on. Such values are larger than one, so they cannot be true correlations. Are they chi-square statistics?  Please, clarify and label accordingly. Being dichotomous variables, a correlation between them is equivalent to the phi coefficient (I believe).

Considering the test of the moderation hypothesis, I would love the authors to report (in the text) the delta-R2 that expresses the specific contribution of the interaction term “Flexibility x Fear of failure” (the delta-R2 = .12 presented in Table 3 included both the interaction term effect and the main Flexibility effect). This will allow a better evaluation of the strength of the moderator effect.

Line 309 – Please, replace the comma before “Taken together (…)”

Line 358 – Please, delete the dot in “[52].can”

Line 478 – The bibliographic reference “Individual differences in statistics anxiety: the roles of perfectionism, procrastination and trait anxiety” should be corrected (the first author is missing).

Line 488 – Please, edit this reference (Ramzi & Saed, 2019). The capital letter for the journal-title and a comma are missing.

Line 492 – Please, check the Pychyl and Sioirs (2016) reference – it seems incomplete.

Line 513 – Please, check: Schmdt et al., 2007 reference is duplicated here.

Line 426-427 – Instead of “plays a key role in negative perfectionism, which leads to procrastination and ego depletion” I suggest “plays a key role in negative perfectionism, which leads to ego depletion and procrastination” (just to maintain the variable sequence in the mediational chain).

Line 429 – Please, consider revising the sentence: “cognitive flexibility can significantly reduce a person's fear of failure” since what has been tested was the moderating effect of cognitive flexibility on the “fear -> ego depletion” path.

What is the meaning of the [J] at the end of the title of each bibliographic reference?

There is no indication of DOI in any bibliographic reference. Is this imposed by the journal Edition style? I found DOI very useful.

Round 2

Reviewer 1 Report

Thank for comprehensive and clear answer. I highly reccomend to publish the article.

Reviewer 2 Report

Good job! The authors have solved my suggestions well.